# Evidence for Multilevel Chemopreventive Activities of Natural Phenols from Functional Genomic Studies of Curcumin, Resveratrol, Genistein, Quercetin, and Luteolin

**DOI:** 10.3390/ijms232314957

**Published:** 2022-11-29

**Authors:** Lukasz Huminiecki

**Affiliations:** Bioinformatics Team, Department of Molecular Biology, Institute of Genetics and Animal Biotechnology of the Polish Academy of Sciences, ul. Postępu 36A, Jastrzębiec, 05-552 Magdalenka, Poland; l.huminiecki@ighz.pl

**Keywords:** chemoprevention, cancer, curcumin, resveratrol, genistein, quercetin, luteolin, genomics

## Abstract

Herein, I present an updated and contextualized literature review of functional genomic studies of natural phenols in the context of cancer. I suggest multilevel chemopreventive and anticancer mechanisms of action, which are shared by multiple dietary natural phenols. Specifically, I cite evidence that curcumin and resveratrol have multilevel anti-cancer effects through: (1) inducing either p53-dependent or p53-independent apoptosis in cancer cell lines, (2) acting as potent regulators of expression of oncogenic and anti-oncogenic microRNAs, and (3) inducing complex epigenetic changes that can switch off oncogenes/switch on anti-oncogenes. There is no simple reductionist explanation for anti-cancer effects of curcumin and resveratrol. More generally, multilevel models of chemoprevention are suggested for related natural phenols and flavonoids such as genistein, quercetin, or luteolin.

## 1. Introduction

### 1.1. Curcumin, Resveratrol, Genistein, Quercetin, and Luteolin Are Small Natural Phenols That Are Important for Chemoprevention

There is a lot of interest in dietary chemoprevention as cancer treatment remains challenging [1]. Dietary chemoprevention is the use of plant-based diet, vitamins, or dietary supplements to reduce the risk or delay cancer development [2]. It is already proven that a diet rich in fruits and vegetables is likely to reduce the risk of cancer in the gastrointestinal tract [3]. In fact, the World Health Organization promotes a diet that includes a minimum of 400 g of fruit and vegetables a day [4]. This is because fruits and vegetables contain antioxidants, for example natural phenols, that have an impact on human cells and tissues [5]. Indeed, natural phenols should be a part of a healthy diet that has epigenetic effects [6].

Note that natural phenols are a widespread and abundant group of plant secondary metabolites. Indeed, as much as 20–30% of all carbon fixed by photosynthesis in plants is directed to the phenylpropanoid metabolic pathway (which is a starting point for the synthesis of natural phenols). Consequently, plant-based foods are the primary source of natural phenols in the human diet. For example, curcumin [7] can be found in the rhizomes of the plant *Curcuma longa*, from which a common spice known as turmeric is derived (turmeric is prized in cooking because of its golden yellow color and exceptionally rich warm and earthy aroma.) Resveratrol is well known to be present in red wine being derived from grape skins. Resveratrol can also be found in many nuts and other fruits.

I chose to focus on curcumin, resveratrol, genistein, quercetin, and luteolin as they are representative small-molecule natural phenols for which there is a significant number of functional genomic datasets in bioinformatics databases. There is also a lot of academic and practical interest in these phytochemicals in the context of chemoprevention (Table 1).

### 1.2. Chemical and Biochemical Activities of the Natural Phenols

Curcumin, resveratrol, genistein, quercetin, and luteolin are the small-molecule natural phenols with chemical structures that are shown in Figure 1. Basic chemical information about these compounds is provided in Table 2.

Phenolic groups of natural phenols can act as hydrogen donors terminating chain reactions of free radicals. Specifically, a chain reaction can end when the phenolic group forms a free radical that is strongly stabilized by conjugation, hydrogen bonds, or delocalization of π-electrons of the benzene ring. Indeed, a detailed computational study [14] suggested that *trans*-resveratrol has antioxidant activities toward both hydroxyl (^•^OH) and hydroperoxyl (^•^OOH) radicals. Four possible mechanisms were demonstrated to be feasible: hydrogen atom transfer—HAT, proton-coupled electron transfer—PCET, sequential electron proton transfer—SEPT, or radical adduct formation—RAF [14]. Note that such computational studies have already been verified experimentally. For example, in vitro assays confirm the antioxidant properties of resveratrol [15]. Likewise, curcumin’s three hydroxyl groups can undergo oxidation by hydrogen abstraction or electron transfer [8]. In another example, a study of liposome oxidation demonstrated the radical-scavenging activity of both trans-resveratrol’s para-hydroxyl and meta-hydroxyl groups [16].

Further note that direct chemical reactions with proteins are also possible through Michael addition products with protein thiols and selenols [16]. For example, curcumin was also shown to spontaneously form Michael adducts with thiol-containing peptides such as glutathione [17]. Moreover, electrophilic Michael acceptors within natural phenols can react with cysteine residues of Kelch-like ECH-associated protein 1 (Keap1) activating the Keap1 NF-E2-related factor 2 (Nrf2) antioxidant response elements (ARE) pathway [18,19]. The Nrf2-ARE pathway regulates from 1% to 10% of the mammalian genes and is integrated with the redox stress signaling system.

It is thus clear that many enzymes and signaling proteins are direct biochemical targets of natural phenols [20,21,22,23]. However, I concentrate herein on the interpretation of transcriptional changes induced by the natural phenols [24,25,26,27]. I argue that gene expression changes induced by natural phenols can be studied using integrative approaches such as functional genomics and bioinformatics (Figure 2).

### 1.3. Carcinogenesis

The process of the development of cancer proceeds in multiple stages over many decades [28]. Consequently, most cancers develop in late adulthood and age is a surprisingly large risk factor for cancer diagnosis. This is almost universally true with the exception of childhood cancers, which typically have a hereditary component. Note that cancers can develop in most normal human tissues. Further note that tissues of cancer origin tend to be those in which primary tumors are identified by clinicians when a symptomatic disease develops.

In clinical practice, tumors are curable at early stages and there are no clinical symptoms beyond localized hyperplasia. However, tumors that are already malignant tend to metastasize beyond in situ lesions invading surrounding tissues. Incurable malignant tumors seed distant metastases in susceptible organs. In final stages, advanced metastatic disease induces biochemical crises (e.g., hypercalcaemia, hyponatraemia, hyperkalaemia, and hyper- or hypoglycaemia). The direct cause of death tends to be bleeding, or clots, or cardiac emergencies, bone marrow suppression, raised intracranial pressure, or multiple organ failure [1].

The main molecular features of the cancer cell, known as cancer hallmarks, have been already identified using molecular biology [29]. Briefly, the hallmarks are as follows: (i) activated oncogenic signaling stimulating cellular proliferation, (ii) disabling of inhibitory controls of cell-cycle, (iii) replicative immortalization [30], (iv) and decreased levels of programmed cell death (i.e., apoptosis).

More precisely, mutagenes or tumor promoters initially induce genetic or epigenetic activation of cellular (or viral) oncogenes in cancer, which then drive the proliferation of transformed cells [28]. The oncogenes are typically growth factors and hormones (e.g., estrogen, androgens), hormone or cytokine receptors (e.g., estrogen receptor, androgen receptor), or intracellular signal transducers (e.g., Ras). In a parallel step, mechanisms that normally block cellular proliferation are inactivated. These anti-proliferative mechanisms include signaling pathways that act through protein tumor suppressors such as phosphoprotein p53 or the retinoblastoma protein—pRB. Additionally, the anti-proliferative mechanisms include apoptosis. In other words, cancer is a disease of aberrant cell signaling with deregulated apoptosis, which could potentially be therapeutically re-activated [31].

It is also worth mentioning that micro-tumors lack the ability to degrade basal membranes, to induce angiogenesis, or to infiltrate surrounding tissues [28]. To progress beyond the size of a few millimeters, tumors must develop new blood vessels and promote the degradation of the extracellular matrix. Moreover, chronic inflammation at the site of the primary tumor and the resulting oxidative stress further promote cancer development. Note that angiogenesis is the process of the development of new blood vessels from old ones. Further note that cancer-inducing inflammation is frequently a result of a chronic viral infection, although the autoimmune process or environmental irritants may also contribute.

### 1.4. Apoptosis

Programmed cell death (i.e., apoptosis) is a process of cell death resulting from the activation of an evolutionarily conserved intracellular “suicide” program. Apoptosis is regulated by a set of critical control points [32]. There are two alternative pathways in which the process of apoptosis can proceed: an intrinsic pathway [33]—linked with mitochondria, or the extrinsic pathway—linked with extracellular signaling (Figure 3).

Apoptosis is also common during animal development [39], in high turnover tissues, or during tissue regeneration. In particular, apoptosis plays a key role during the development of vertebrates. In this case, an expanded molecular machinery of the apoptotic pathway [40,41] shapes a complex vertebrate body plan and sculpts fine body structures such as fingers or brain compartments. Adult animals must also repair or eliminate damaged or malfunctioning cells from their differentiated tissues, ensuring that organs continue to fulfill their physiological functions. For example, billions of epithelial cells die every hour in bone marrow and intestine. Programmed cell death is also essential for the maintenance of the immune system [42].

Note that decreased apoptosis is well known to be one of the major hallmarks of the cancer cell (Hanahan and Weinberg 2011). In a manner of speaking, cancer represents a failure of normal control mechanisms to induce apoptosis in an aberrant cell lineage. Further note that curcumin and resveratrol can induce apoptosis in cancer cell lines by various mechanisms and at relatively low concentrations [43,44,45,46]. This effect is specific to transformed cells, i.e., cells that have mutated genomes and/or aberrant expression profiles.

## 2. Results of Functional Genomic Studies of Natural Phenols

### 2.1. Natural Phenols Induce Apoptosis in Cancer Cell Lines in a P53-Dependent Manner

Apoptosis induced by natural phenols in cancer cells could proceed through either of the following three mechanisms: (1) activation of the p53 pathway; (2) re-activation of the p53 pathway; and (3) through one of p53-independent pathways. Importantly, published functional genomic screens found the anti-cancer activity of resveratrol (Table 3) and curcumin (Table 4) to be mediated by the intrinsic apoptosis pathway. Seven out of eight studies found explicit evidence for the activation of the p53-dependant apoptosis pathway. This evidence is important, as functional genomic screens are unbiased, measuring expression levels of all the relevant genes at the same time. Technically, the activation of p53-dependant apoptosis was inferred from transcriptional upregulation of p21, or p27, or MDM2 in a microarray dataset (in other words, p53-inducible genes were present among differentially expressed genes—DEGs).

However, out of the studies listed in Table 1 and Table 2 only Chin et al. used a p53-mutated cell line. It was MDA-MB-231—a breast cancer cell line that is known to harbor a p53 mutant stabilized by elevated phospholipase D activity [55]. To put that in context, it is well known that over 50% of cancers have loss-of-function mutations in *p53* [56]. At the same time, probably all cancer cell lines inactivate the p53 pathway by other genetic or epigenetic mechanisms. For example, this can happen by mutations in one of the other pathway components than p53 or by epigenetic changes within genes that code for pathway components or small RNA regulators.

### 2.2. Natural Phenols as Potent Regulators of Expression of Oncogenic and Anti-Oncogenic microRNAs

Soon after the discovery of microRNAs—miRNAs [57]—it was recognized that they can be components of both oncogene and anti-oncogene pathways [58]. For example, it was proven that the p53 transcription factor regulates expression of several microRNAs of the miR-34 family, which consists of three microRNAs: miR-34a, miR-34b, and miR-34c [59]. MiR-34a, in particular, has several well-characterized tumor suppressor functions in a variety of tumors. For example, miR-34a down-regulates the expression of oncogenes such as c-Myc, c-Met, c-Kit, androgen receptor; anti-apoptotic proteins such as Bcl2; and MDM4—a major negative regulator of p53 [59]. More generally, oncogenic miRNAs could regulate or buffer expression of hundreds of target genes involved in apoptosis, cell cycle arrest, or senescence [60,61]. Finally, mutations of microRNA genes themselves, as well as of their targets, of processing proteins, or of their epigenetic modifications, are now recognized to have a broad role in carcinogenesis [62].

General reviews suggest that phytochemicals [63], polyphenols [64], and specifically curcumin [65] might modulate various cancer-related cellular processes, such as proliferation, apoptosis, and inflammation by affecting microRNA activities. For example, curcumin was shown to reverse the Bisphenol A-induced upregulation of oncogenic miR-19a and miR-19b in estrogen-receptor-positive MCF-7 human breast cancer cells [66]. Moreover, in colon cancer cells where curcumin induces reactive oxygen species (ROS), curcumin was found to down-regulate microRNAs miR-27a, miR-20a, and miR-17-5p and regulate transcriptional repressors ZBTB10 and ZBTB4 [67]. Another paper suggested that ROS induced by curcumin can reverse drug resistance phenotypes in colon cancer cells by the suppression of microRNA-27a and induction of ZBTB10 [68]. The most recent review by Akbari et al. [69] suggested that curcumin modulates microRNAs in ways that moderate oxidative stress and enhance apoptosis in gastric cancer, colorectal cancer, hepatocellular cancer, pancreatic cancer, and esophageal cancer.

Were the effects of polyphenols on microRNAs confirmed by unbiased functional genomic screens? Indeed, an early study demonstrated that curcumin significantly alters expression profiles of many microRNAs in human pancreatic cancer cells [70]. Eleven microRNAs were up-regulated and eighteen microRNAs were down-regulated, thus suggesting broad changes in the complex regulatory network. In particular, miRNA-22 was up-regulated by 65.5% and miRNA-199a* was down-regulated by 54.2%. Among the identified target genes of miRNA-22, there were many oncogenes such as estrogen receptor 1 (ESR1), Rap guanine nucleotide exchange factor (RAPGEFL1), the RAB5B member of the RAS oncogene family, TYRO3 protein tyrosine kinase (TYRO3), FOS-like antigen 1 (FOSL1), fibroblast growth factor receptor 2 (FGFR2), and mitogen-activated protein kinase kinase kinase 3 (MAP3K3).

Moreover, Ye et al. [54] suggested that curcumin may also regulate anti-oncogenes through complex microRNA regulatory networks. The study demonstrated that curcumin promotes apoptosis in non-small cell lung cancer through the p53-miR-192-5p/215 pathway. This effect depends on the activation of p53. In the second step, p53 up-regulates miR-192-5p and miR-215. These microRNAs in turn down-regulate the X-linked inhibitor of apoptosis (XIAP). XIAP is a member of a family of structurally related proteins that are inhibitors of caspase family cell death proteases [71].

Broad changes in microRNA expression were also reported in the studies of resveratrol by MiRNA arrays. For example, Bae et al. [72] examined how resveratrol affected microRNA expression profiles in non-small cell lung cancer: there were as many as 71 miRNAs exhibiting greater than 2-fold expression changes in resveratrol-treated cells. The changes were more than 20-fold for miR-299-5p, miR-194*, miR-338-3p, miR-758, miR-582-3p, and mir-92a-2*. The authors concluded that changes of expression of microRNAs may underlie resveratrol’s chemoprotective effects against human lung cancer.

There was also an in vivo study of metastatic adenocarcinoma deriving from prostate [73]. Fifteen miRNAs were confirmed to be down-regulated and ten miRNAs were up-regulated. Note that the MiRNA array detected broad changes in microRNA expression, but the authors decided to focus only on those microRNAs that were previously shown to be regulated in prostate cancer [74]. The authors suggested that resveratrol might inhibit tumor growth and invasiveness through the inhibition of the expression of one of the pre-selected microRNAs: the oncogenic miR-21. Crucially, miR-21 regulates a number of antiapoptotic genes [75]. Moreover, miR-21 is modulated by the Akt signaling pathway [76].

### 2.3. Natural Phenols as Dietary Epidrugs

In 2010, cancer chemoprevention by dietary polyphenols relies on epigenetic effects such as changes in microRNA expression, histone modifications, or DNA methylation changes [77]. Subsequently, Link et al. presented a comprehensive and systematic dataset consisting of the integrated expression and methylation data for colorectal cancer cell lines incubated in vitro with curcumin [78]. Specifically, the authors integrated data from high-throughput microarrays profiling expressions of 25 thousand genes and microarrays analyzing the promoter methylation status of 26 thousand individual CpG sites over fourteen thousand genes [79]. The cell lines were treated with curcumin for either six days or eight months.

Note that resveratrol could also induce widespread methylation remodeling in a breast cancer cell line after nine days [80]. The nine-day treatment also suggests that epigenetic reprogramming was secondary to modulation of signaling pathways and expression changes. Further note that the actions of resveratrol were reinterpreted as activities of a sort of “dietary epidrug” that induces widespread but specific epigenetic changes [81]. In fact, it was suggested that changes in DNA methylation changes in DNA isolated from blood cells could be used as a sort of quantitative measure of long-term dietary exposure to dietary polyphenols [82]. Moreover, dietary polyphenols could be helpful in reversing pathological methylations states that characterize chronic diseases [83] or cancer.

### 2.4. Epigenetic Changes Induced by Natural Phenols in Cancer Cells Switch off Oncogenes and Switch on Anti-Oncogenes

Almost ten years ago, Link et al. [78] detected a significant correlation between DNA methylation and gene expression changes in cells subjected to long-term curcumin treatment. In this particular experiment, methylation changes appeared independent of DNA methyl transferases (DNMTs). That is to say, no strong changes in expression or activity of the de novo DNA methyltransferase DNMT1 were reported; moreover, curcumin-induced methylation changes were limited and specific (unlike gross methylation changes in LINE-1 repeat elements that could be attributed to strong DNMT inhibitors). Importantly, significant expression changes were detected in the components, interactors, and transcriptional targets of the NF-κB pathway.

More recently, Beetch et al. [84] performed a detailed study to identify the mechanism of methylation changes induced by natural phenols in vitro in cancer cells. Specifically, MCF10CA1a breast cancer cells were treated for four or nine days in a 15 µm solution of resveratrol and characterized using molecular biology and methylation microarrays (Illumina Infinium Human Methylation 450K BeadChip). Importantly, methylation changes induced upon exposure to resveratrol were linked to decreased DNMT3A expression, as measured by qPCR. Note that DNA methyltransferase 3A—DNMT3A is a de novo methyltransferase involved in establishing DNA methylation patterns in development and cancer [85]. In summary, most epigenetic changes detected in the cancer cells after in vitro incubation with resveratrol were specific and many could be mapped to oncogenic pathways. The genes identified included cytokine receptors, oncogenic transcription factors, or epigenetic regulators.

More specifically, the authors of the above study focusing on resveratrol suggested that there is a set of genes (116 CpG sites were listed) that were preferentially hypo-methylated upon exposure to resveratrol in breast cancer cell lines in comparison to normal breast epithelium. The list of CpG sites included promoters and 5′UTRs of AGTPBP1, SEMA3A, FOXN3, UACA, FAM49A, TMEM91, CSMD1, WFDC3, EPN2, and HIST1H2BK. Even more specifically, the authors focused on the SEMA3A locus. There was decreased binding of DNMT3A within the semaphorin-3A (SEMA3A) promoter, followed by increased nuclear factor 1 C-type (NF1C) occupancy in this locus.

At the same time, in an in vitro study of curcumin in several human cancer cell lines, curcumin induced some global DNA hypomethylation and altered the expression of three DNMTs: DNMT1, DNMT3A, and DNMT3B [86]. However, such global changes were accompanied by changes that were specific. For example, the p21 proximal promoter was de-methylated in the curcumin-treatment group. Curcumin also increased the KLF4 and p53 expression and increased KLF4 occupancy within the proximal promoter of p21.

In an alternative in vivo model of colon cancer accelerated by inflammation, curcumin induced specific rather than general changes in genomic methylation patterns [87]. Specifically, dietary curcumin induced alterations in the DNA methylome and transcriptome, which could be mapped to several cancer-associated signaling pathways as well as to genes important for inflammatory responses. For example, curcumin could reverse the hypomethylation status of the tumor necrosis factor.

Taken together, the studies listed in this section suggest that methylation changes induced by polyphenols are multilevel but tend to be specific rather than global.

### 2.5. Similar Results Can Be Obtained for Other Natural Phenols

In the preceding sections, I performed a deliberately paced review of high-quality functional genomic datasets available for curcumin and resveratrol (Table 5). It was my main aim to provide supporting evidence for a model of multilevel effects of the two natural phenols. I listed functional genomics studies, one by one, which provide support for individual mechanisms of action (Section 2.1, Section 2.2, Section 2.3, Section 2.4). In Section 2.5.1, Section 2.5.2, Section 2.5.3, I want to suggest that generally similar results can be obtained for related phytochemicals such as flavonoids (Table 6).

#### 2.5.1. Genistein

Genistein is a redox active isoflavonoid [11]—a type of natural phenol—that is also biologically active as a phytoestrogen and is abundant in soybeans. In cancer cells, this phytochemical has anti-proliferative properties, recognized especially in cases of breast cancer [98]. It is suspected that genistein prevents angiogenesis, inhibits tyrosine kinases, modulates the hedgehog signaling pathway, and induces complex epigenetic changes. In in vitro studies on cancer cell lines, genistein was shown to induce cell cycle arrest and apoptosis; interfering with many signaling pathways, as well as with the expression of both coding and non-coding genes [98,99].

Functional genomic studies of genistein confirmed that this compound can induce apoptosis in tumors in vivo and that it inhibits many signaling pathways important for prostate cancer progression [88]. Specifically, Bilir et al. determined that genistein modulated the expression of genes involved in angiogenesis, epithelial-to-mesenchymal transition, and PDGF binding. Moreover, there was also a set of differentially methylated genes, such as genes involved in developmental processes, stem cell markers, or genes involved in proliferation and transcriptional regulation.

Another microarray study of genistein [89] identified a set of signaling pathways that could be involved in dietary protection from chemically induced mammary cancer. The DEGs clustered into functional pathways such as metabolic control, immune response, signal transduction, growth regulation, and ion transport.

Finally, Chiyomaru et al. [90] examined the impact of in vitro genistein treatment on the expression of miRNAs and *lnc*RNAs in prostate cancer cell lines. The authors underlined that genistein slowed down the proliferation of cancerous cells specifically by down-regulating oncogenic *lnc*RNA HOTAIR.

At the same time, a cancer clinical trial of high soy intake suggested that high plasma levels of genistein, and its inactive form daidzein [100], can speed up breast cancer growth in vivo [92]. In this study, the rate of cancer cell proliferation was increased in biopsies derived from a high soy group versus control. However, this effect is likely to be specific to some types of breast cancer; this is because as many as 39 out of 51 resected tumor specimens were estrogen receptor positive (recall that genistein is considered a potent phytoestrogen).

Finally, there is a report listing differentially expressed long non-coding RNAs in daidzein-treated lung cancer cells [93]. Although daidzein is an inactive analog of genistein, it also has antioxidant and phytoestrogenic properties. Following the addition of daidzein, the expression level of eight long non-coding RNAs was up-regulated and 111 were down-regulated. These numbers were higher than those for protein-coding genes, as only five mRNAs were up-regulated and 35 mRNAs were down-regulated.

#### 2.5.2. Quercetin

There is molecular evidence for similarly complex multilevel effects in case of quercetin [12]—a polyphenolic flavonoid that can also have chemopreventive effects [101]. For example, quercetin can induce widespread changes in gene expression in epithelial cells by modifying the activity or expression of many transcription factors [94]. The differentially regulated transcripts mapped functionally to pathways such as cell cycle, cell adhesion, gene transcription, xenobiotic metabolism, and immune responses.

Another study profiled both the transcriptome and the proteome of colon mucosa in rats fed with a diet supplemented in quercetin [95]. Quercetin, at the dose of 10 g per kilogram given for eleven weeks, down-regulated oncogenic pathways, for example the mitogen-activated protein kinase pathway. The phytochemical also increased the expression of tumor suppressor genes such as phosphatase and tensin homolog (Pten), p53 or DNA mismatch repair protein MutS homolog 2 (Msh2), as well as cell cycle inhibitors. Note that proteomics revealed fewer differentially affected proteins and there was a low correlation between changes in gene expression and changes in protein levels.

However, another study demonstrated that quercetin can induce changes in gene expression that could directly lead to increased protection from free radicals [91]. In particular, the phytochemical induced changes in the expression of genes involved in lipid metabolism, such as stearoyl-CoA-desaturase 1, 3-hydroxy-3-methylglutaryl-coenzyme A (HMG-CoA) reductase, and sestrin 1. This was most likely due to the elevated expression and activation of sterol regulatory element-binding protein-2 (SREBP-2).

#### 2.5.3. Luteolin

Irreducibly complex transcriptomic evidence was also reported for luteolin [13]—another dietary flavone found in many plant foods [96]. Specifically, Sakurai et al. profiled a metastatic prostate cancer cell line—PC3—incubated with either gefitinib or luteolin. The study reported that both gefitinib and luteolin cause growth arrest of cancer cells. The effects were mechanistically multilevel, involving the modulation of the activities of enzymes (such as Ser/Thr kinase activity of cyclin G-associated kinase), and changes in the expression of microRNAs (such as miR-630 and miR-5703).

In a mouse xenograft model of head and neck squamous cell carcinoma, luteolin reduced tumor growth [97]. The effects of luteolin were again at multiple levels, including the modulation of gene expression, miRNA expression, and processing of miRNAs. Interestingly, luteolin was found to inhibit p300 acetyltransferase activity with competitive binding to the acetyl CoA binding site. The inhibition of p300 acetyltransferase led to the alteration of expression profiles of many genes, including tumor protein p53 (p53), miR-195/215, let7C, or the miR-135a oncomiRNA. As many as 40 miRNAs showed more than 2-fold differential expression.

## 3. Conclusions

In general, preceding laboratory observations of apoptosis induced by a redox-active compound in some cell line model later motivated follow-up functional genomic studies. In other words, molecular biologists typically first discovered that cancer cells stopped to proliferate when exposed to curcumin or resveratrol. Only later, they looked for DEGs induced by the treatment with a polyphenol using microarrays or RNAseq [25,27]. I argue for a multilevel interpretation of the genomic studies, stressing that there is no simple reductionist explanation for all anti-cancer effects of natural phenols.

The effects of natural phenols on cancer cells are to oppose homeostasis and proliferation. Specifically, I showed in results sections that natural phenols restore normal the tendency toward apoptosis in cancer cells (Section 2.1), that they act as potent regulators of expression of oncogenic and anti-oncogenic microRNAs (Section 2.2), that they induce complex epigenetic changes (Section 2.3), and switch off oncogenes and switch on anti-oncogenes (Section 2.4).

As a modification of the multilevel model, I propose to identify two stages of responses to natural phenols. For example, treatment with curcumin first interferes with protein components of signaling pathways. Subsequently, transcriptional factors regulated by signaling pathways modify the expression of early and late transcriptional targets [27]. Only then there are long-term specific methylation changes. In other words, curcumin directly interferes with signaling pathways and it indirectly modifies gene expression. Finally, there are following long-term epigenetic changes.

Finally, note that studies reviewed herein do not provide evidence supporting sirtuins or Nrf2 as main mediators of responses to natural phenols. Instead, the evidence reviewed here favors an alternative explanation that underlines an evolutionary adaptation to high environmental levels of natural phenols. Because of the ubiquity of plant-derived redox active phytochemicals, evolutionary selection probably acted on animal genes in such ways as to buffer their interactions with animal enzymes and signaling pathways. As a consequence, natural phenols have multilevel effects on mammalian cells described in Section 2.1, Section 2.2, Section 2.3, Section 2.4. There are also long-term epigenetic changes coordinated with immediate changes in gene expression levels. According to a similar hypothesis that was previously published, natural phenols induce a defensive state in mammalian cells. In a defensive state, activities and gene expression of many detoxifying and repair enzymes are enhanced by allosteric or covalent modifications [22].

However, note that other phytochemicals can accompany and interact with these simple natural phenols in foods. For example, resveratrol in the skins of grape berries are accompanied by tannins, anthocyanins, catechins, ellagic acid, and flavonids. Curcumin is accompanied in turmeric by carbohydrates, proteins and peptides, minerals, other curcuminoids (demethoxycurcumin, bis-demethoxycurcumin, and others), plant steroids, and essential oils enriched in sesquiterpenes.

Computational analyses of genomic data can lead to multilevel biological interpretations. For example, one can conclude that natural phenols induce the expression of apoptosis-related genes and microRNAs. Moreover, oncogenes are switched off and anti-oncogenes are switched on epigenetically.

Alternatively, an extrinsic apoptosis pathway can be activated by trans-membrane receptors called death receptors [102]. In the second step, the apoptotic signal is amplified through of several alternative mechanisms: a proteolytic cascade, or transcriptional changes, or the release of cytochrome *c*. Finally, cells dying through apoptosis are broken into chemically marked apoptotic bodies. The apoptotic bodies are recognized and neatly eliminated by neighboring cells and macrophages.

Importantly, curcumin- and resveratrol-induced apoptosis proceeds normally through the intrinsic pathway that is p53-dependent. For example, Lin et al. reviewed evidence suggesting that resveratrol induces apoptosis through the p53 pathway, integrin α_v_β_3_, and the mitogen-activated protein kinase—MAPK [103]. Another review pointed to breast cancer as a type of tumor, in which curcumin can induce the intrinsic pathway by simultaneously up-regulating pro-apoptotic proteins such as p53 and Bax and down-regulating antiapoptotic proteins such as MDM2 and Bcl-2 [104].

However, few other reports suggested a few disparate mechanisms that do not seem to require the activation of p53. These mechanisms include apoptosis induced by the tumor necrosis family death receptor [105,106], apoptosis induced by Bax protein oligomerization [107], rapid perinuclear mitochondrial clustering [108], oxidative stress [109], necrosis-like apoptosis [110], or p38 mitogen-activated protein kinase activation and down-regulation of survivin expression and Akt signaling [111]. It seems that these p53-independent mechanisms group into two major classes: (1) the extrinsic apoptosis pathway induced by signaling from death receptors, and (2) necrosis-like apoptosis that is p53-independent but it still involves mitochondria, release of cytochrome C, and activation of caspases.

We chose to focus on curcumin, resveratrol, genistein, quercetin, and luteolin as they are important for cancer chemoprevention. There are also many food supplements available commercially that contain these phytochemicals. The table shows numbers of cancer-related articles and reviews registered in PubMed as of October 2022, as well as a number of relevant clinical trials registered at ClinicalTrials.gov. Moreover, number of food supplements available on Amazon.com “Health and Household” is provided.

## Figures and Tables

**Figure 1 ijms-23-14957-f001:**
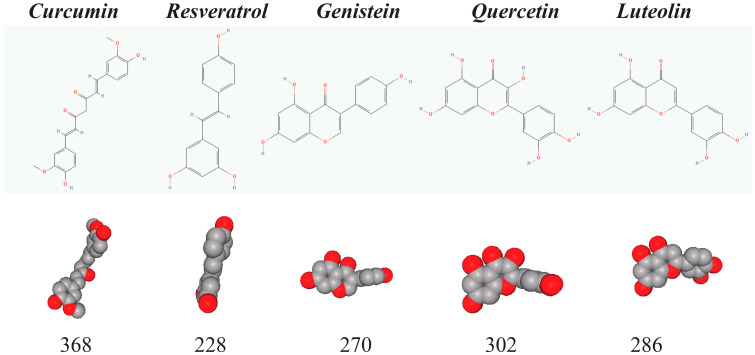
**Chemical structures of curcumin, resveratrol, genistein, quercetin, and luteolin.** This figure shows chemical structures (top row), space-filling models (medium row), and exact mass (bottom row) of curcumin, resveratrol, genistein, quercetin, and luteolin. Curcumin is a diarylheptanoid from the group of curcuminoids, while resveratrol is a 3,4′,5-trihydroxy-trans-stilbene. Genistein, quercetin, and luteolin are flavonoids derived from a 15-carbon skeleton with two aromatic phenyl rings (A and B). Flavonoids also contain a heterocyclic pyran ring (C) with an embedded oxygen. These are small molecules—with molecular weight between 228 and 368—that are aromatic. For example, curcumin and resveratrol contain two aromatic phenyl rings. Note that such natural phenols are chemically active in many types of organic reactions that are common in a living cell, such as free radical chemistry, nucleophilic addition, or metal ion interactions [8].

**Figure 2 ijms-23-14957-f002:**
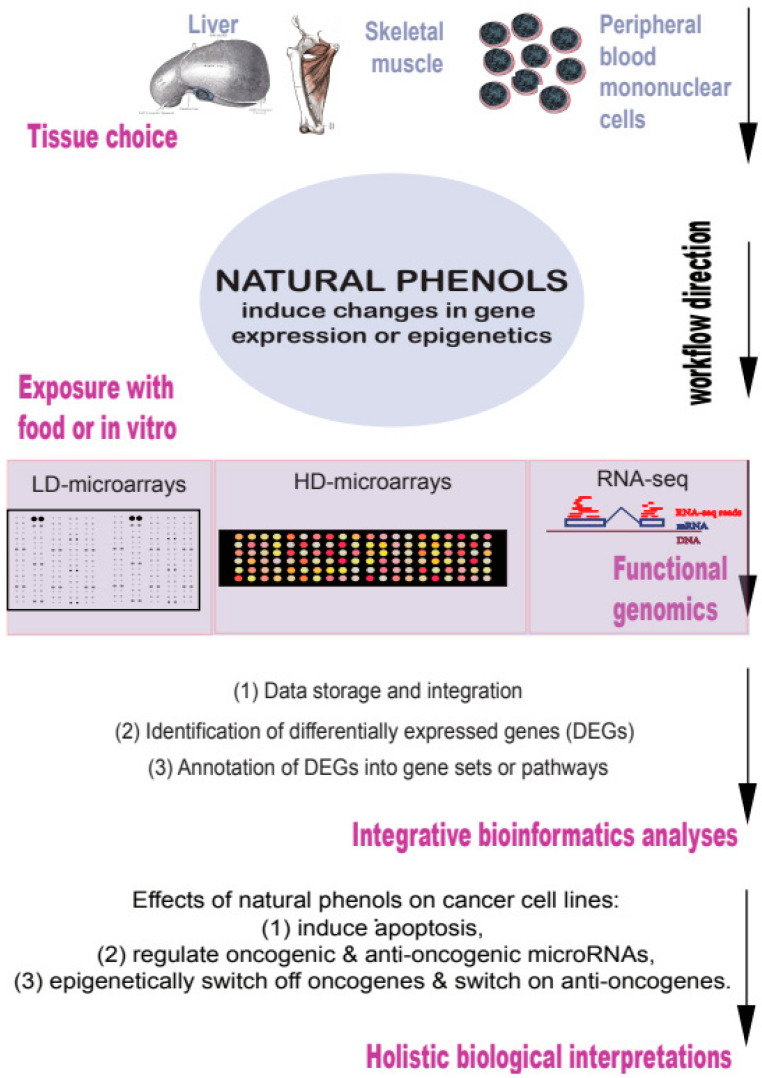
Natural phenols induce complex gene expression changes that can be studied using functional genomics and anti-reductionist bioinformatics.

**Figure 3 ijms-23-14957-f003:**
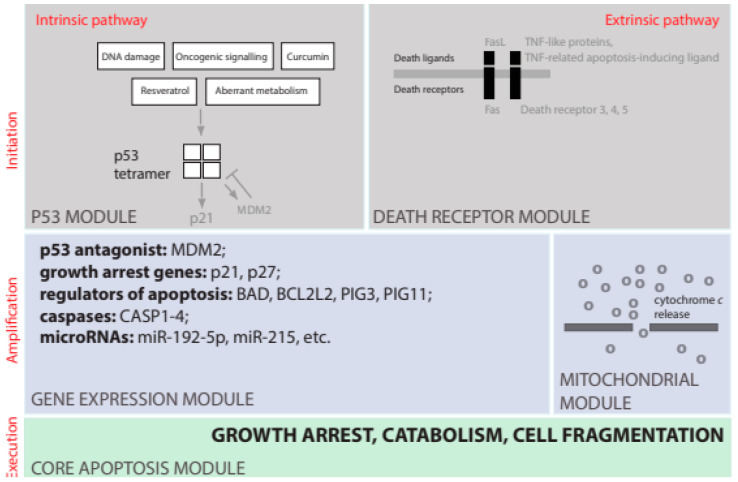
**A circuit board for initiation, amplification, and execution of apoptosis.** Apoptosis is initiated through either a p53-dependent (intrinsic) or independent (extrinsic) pathway. The intrinsic apoptosis pathway is p53-dependent [34] and is initiated by pathological internal cellular states such as DNA damage or aberrant metabolism. Cyclin-dependent kinase inhibitor 1—p21 [35] and cyclin-dependent kinase inhibitor—p27 [36] are key inhibitors of the cell cycle that is transcriptionally induced by p53. Interestingly, mutated p53 is frequently stabilized by mutations rather than depleted in cancer cells [37]. This is because normal p53 functions within a dynamic cycle of production and degradation known as the *futile cycle*. Indeed, mutated p53 forms lack normal transcriptional activity and fail to transcriptionally induce MDM2—a protein that induces rapid p53 ubiquitination [38].

**Table 1 ijms-23-14957-t001:** There is a lot of interest in curcumin, resveratrol, genistein, quercetin, and luteolin in the context of cancer.

Compound	Articles	Reviews	Clinical Trials	Supplements
Curcumin	7157	1324	86	318
Resveratrol	4393	1067	22	214
Genistein	3474	489	33	15
Quercetin	4019	509	20	310
Luteolin	1048	102	3	100

**Table 2 ijms-23-14957-t002:** Chemical information.

Compound	Synonyms	Reactive Group	PubChem * Reference	PubChem ID	Molecular Formula
Curcumin	Diferuloylmethane	Ether, ketone, phenol, unsaturated aliphatic hydrocarbon, hydroxyl	[9]	969516	C_21_H_20_O_6_
Resveratrol	3, 4′, 5-Trihydroxystilbene	Stilbene, phenol, hydroxyl	[10]	445154	C_14_H_12_O_3_
Genistein	4′, 5, 7-Trihydroxyisoflavone	Phenol, hydroxyl, ketone	[11]	5280961	C_15_H_10_O_5_
Quercetin	3, 3′, 4′, 5, 7-Pentahydroxyflavone	Phenol, hydroxyl, ketone	[12]	5280343	C_15_H_10_O_7_
Luteolin	3′, 4′, 5, 7-Tetrahydroxyflavone	Phenol, hydroxyl, ketone	[13]	5280445	C_15_H_10_O_6_

* Open chemistry database.

**Table 3 ijms-23-14957-t003:** There is strong evidence for the activation of the intrinsic apoptosis pathway by resveratrol.

Technological Platform	Cell Line and Treatment Conditions	Reported Mechanism	Other Conclusions	Reference
LD-microarray profiling expression of coding genes.	LNCaP human prostate adenocarcinoma cells (androgen-sensitive, p53 wt/wt).	Resveratrol induced the intrinsic apoptosis pathway (but only at the highest concentration tested).	At intermediate concentrations, resveratrol modulated cell cycle regulatory genes, and down-regulated markers of cellular proliferation. (At the lowest concentrations, resveratrol stimulated viability suggesting a hormetic response curve.)	[47]
	The phytochemical was applied in cell media in increasing concentrations: 0.01, 0.1, 1, 10, 25, 40, and 100 μM.	The activation of p53-dependant apoptosis was evident by the transcriptional upregulation of p21 and *MDM2*.	Resveratrol also had a strong inhibitory effect on the androgen pathway (which included prostate-specific antigen—PSA).	
HD-microarray profiling expression of coding genes.	The following human lung cancer cells were used: NCI H460 (p53 wt/wt), NCI H23 (with a homozygous missense mutation: methionine to isoleucine at codon 246), and A549 (p53 wt/wt).	Resveratrol induced the intrinsic apoptosis pathway in wild-type A549 cells.	Resveratrol also inhibited growth of the p53 mutated cancer cell line (NCI H23) suggesting that it can also induce p53-independent apoptosis or cell cycle arrest.	[48]
	25 μM phytochemical was applied in cell media (with the incubation time of 48 h).	The activation of p53-dependant apoptosis was evident by transcriptional upregulation of p21 and p27.		
	Human fibrosarcoma cells: HT1080 (p53 wt/wt).	Resveratrol induced the intrinsic apoptosis pathway.	Resveratrol also modulated the expression of genes associated with cell cycle, cytoskeleton, and cell-adhesion.	[49]
	The phytochemical was applied at 2195 ng/mL in media (with the incubation time of six hours).	The activation of p53-dependant apoptosis was evident by differential regulation of 13 genes in the KEGG’s p53 signaling pathway.		
	MDA-MB-231 human breast cancer cell line (estrogen receptor negative).	Resveratrol induced the intrinsic apoptosis pathway.	There was also evidence for cell cycle arrest (increased fraction of cells in the G_1_ phase and inhibition of the expression of cyclin B1).	[50]
	10 μM phytochemical was applied in cell media (with the incubation time of six hours).	The activation of the p53-dependant apoptosis was evident by transcriptional upregulation of p21, *PIG3*, and *BAD*.		
	Human renal carcinoma cells of likely proximal tubule origin. (These cells are known to have a p53 wt/wt genotype, but p53 signaling is strongly repressed.)	Resveratrol induced the intrinsic apoptosis pathway.	Fifteen-fold induction of tumor necrosis factor α inducible protein 3 (TNFAIP3) also suggested the induction of the extrinsic apoptosis pathway.	[51]
	50 μM phytochemical was applied in cell media (with the incubation time of 24 h).	The activation of the p53-dependant apoptosis was evident by sevenfold transcriptional upregulation of *MDM2*.		

**Table 4 ijms-23-14957-t004:** There is strong evidence for the activation of the intrinsic apoptosis pathway by curcumin.

Technological Platform	Cell Line and Treatment Conditions	Reported Mechanism	Other Conclusions	Reference
LD-microarray profiling expression of coding genes.	MCF-7 human breast adenocarcinoma (p53 wt/wt).	Resveratrol induced the intrinsic apoptosis pathway.	Changes in expression of some apoptosis-related genes were dependent on the concentration of curcumin: shifting in opposite directions at the low versus the high concentration. (This might suggest hormetic effects.)	[52]
		The activation of the p53-dependant apoptosis was evident by transcriptional up-regulation of CASP1, CASP2, CASP3, CASP4, BCL2L2, PIG3, and PIG11.		
	Cells were incubated in media with 0, 25, or 50 μg/mL phytochemical (applied for 24 h).			
		Observed changes also suggested indirect activation of apoptosis by down-regulation of pro-survival signals from growth factors and apoptosis inhibitors.		
HD-microarray profiling expression of coding genes.	Hepatocellular carcinoma cell lines: KMCH, WRL68, Huh7, PLC, and Pitts1.	Curcumin indirectly promoted apoptosis by silencing pro-survival oncogenic signaling, in particular the NF-κB pathway.	Curcumin also modulated MYC signaling, which is known to have mixed effects (partially mitogenic and partially pro-apoptotic). Curcumin also modulated genes involved in cytokine signaling, growth factor signaling, and the regulation of angiogenesis.	[53]
	Cells were incubated with 25 μM phytochemical (applied for 72 h).			
A microarray profiling *micro*RNAs (miRCURY).	Several human cell lines originating from non-small cell lung cancers.	Resveratrol induced the intrinsic apoptosis pathway.	Altogether, six microRNAs were up regulated by the curcumin treatment: miR-132-3p, miR-183-5p, miR-124-3p, miR-215, miR-192-5p, and miR-194-5p. Moreover, two microRNAs were down regulated (i.e., miR-602 and miR-223-3p).	[54]
		The activation of the p53-dependant apoptosis was evident by up-regulation of pro-apoptotic miR-192-5p and miR-215.		
		The microRNAs acts similarly to protein-coding anti-oncogenes inducing apoptosis in transformed cells. The induction of apoptosis involved the down-regulation of a known inhibitor of the intrinsic apoptosis pathway: X-linked inhibitor of apoptosis protein (XIAP—which normally binds and blocks initiator caspase 9). The resulting activation of the initiator caspase, in turn, led to the activation of effector caspases (in particular caspase-3) that rapidly effected cell death.		
	Cells were incubated with 15 μM curcumin for 48 h.			

**Table 5 ijms-23-14957-t005:** Functional genomic studies of resveratrol and curcumin.

Compound	Cancer Model	Chip	Data
Resveratrol	Prostate adenocarcinoma cell line [47]	LD-microarray with 30,000 UniGene clusters	GSE4399
	Non-small cell lung cancer [48]	Human Genome U133 Plus 2.0 Array (Affymetrix)	GDS2966
	Fibrosarcoma cell line [49]	Human Genome U133 Plus 2.0 Array (Affymetrix)	GSE59704
	Breast adenocarcinoma [50]	HD-microarray	n/a
	Kidney carcinoma [51]	LD-microarray	n/a
Curcumin	Breast adenocarcinoma [52]	LD-microarray, 4069 I.M.A.G.E. clones spotted on microscopic slides	n/a
	Liver hepatocellular carcinoma [53]	Human HT-12 V4.0 expression Beadchip (Illumina)	GSE59713
	Non-small cell lung cancer [54]	MicroRNA microarray	n/a

**Table 6 ijms-23-14957-t006:** Functional genomic studies of genistein, quercetin, and luteolin.

Compound	Cancer Model	Chip	Data
Genistein	Prostate specimens from a clinical trial of genistein supplementation prior to prostatectomy [88]	HumanHT-12 v4 Expression BeadChip, HumanMethylation450 (Illumina)	GSE84748, GSE84749
	Rat mammary epithelial cells [89]	Rat 230A GeneChip (Affymetrix)	GSE6879
	Prostate cancer cell line [90]	SurePrint G3 Human GE 8 × 60K Microarray (Agilent)	GSE29079
	Human embryonic kidney cell line—HEK293, and breast cancer cell line—MCF-7 [91]	14K microarray slides printed at the University of Calgary	GSE6199, GSE6200
	Women with invasive breast adenocarcinoma [92]	Human U133 Plus 2.0 chip (Affymetrix)	GSE58792
Daidzein	H1299 lung cancer cells [93]	Capital Bio Technology human long non-coding RNA Array v4	GSE181093
Quercetin	HepG2 hepatocellular carcinoma cell line and breast adenocarcinoma T47D cells [94]	Human U133A Plus 2 chip (Affymetrix)	GSE15162
	Distal colon mucosa of rats [95]	Rat Genome 230 2.0 Array (Affymetrix)	GSE7479
	Neuroblastoma cells in vitro [91]		GSE6200
Luteolin	Prostate cancer cell line [96]	SurePrint Human v18.0 miRNA array, whole human genome oligonucleotide microarrays (Agilent)	GSE53180, GSE53178
	Mouse xenograft model of head and neck squamous cell carcinoma [97]	SurePrint G3 Human GE v2 8 × 60K array, human miRNA microarray (Agilent)	GSE75029

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
