# Peer review of "Evidence for Multilevel Chemopreventive Activities of Natural Phenols from Functional Genomic Studies of Curcumin, Resveratrol, Genistein, Quercetin, and Luteolin"

_ijms, 2022, doi:10.3390/ijms232314957_

Round 1

Reviewer 1 Report

After the minor revision, especially, the change of references in the text e.g. sentences 156, 162, and the others, the paper will be published. Furthermore, you must add some images in accordance with the reference list.

Author Response

Referee 1.
>After the minor revision, especially, the change of references in the text e.g. sentences 156,
>162, and the others, the paper will be published.

Sentences 156 and 162 contained footnotes not references. Text from these footnotes was moved into the main body of the text.

>Furthermore, you must add some images in

>accordance with the reference list.

Chemical structures of curcumin, resveratrol, genistein, quercetin, an
d luteolin are shown in Figure 1. A simple scheme for apoptosis was added as Figure 3.

Reviewer 2 Report

There is Evidence for Holistic Anti-cancer Activities of Natural Phenols in Functional Genomic Studies of Curcumin, Resveratrol, Genistein, Quercetin, and Luteolin.

The subject of this review is very interesting and not very well addressed up to now: How is the combined effect of multiple dietary bioactive compounds on multiple targets in a complex system as a cell? Therefore, I looked very much forward to review the manuscript, but I was honestly disappointed when I realized that the paper is in a what I will call “a preliminary version”.

Major concerns

1.       This is a follow-up on previous reviews from the same author concerning resveratrol and curcumin. It is mostly a repetition of what is already found in the previous reviews (Huminiecki, L. and J. Horbanczuk (2018). "The functional genomic studies of resveratrol in respect to its anti-cancer effects." Biotechnology advances 36(6): 1699-1708. And Huminiecki, L., J. Horbanczuk, et al. (2017). "The functional genomic studies of curcumin." Seminars in cancer biology 46: 107- 647). It is not clear whether new data are included. I don’t think so.

2.       It is stated that “my aim herein is to argue for a holistic and contextualized interpretation of all the genomic studies. In particular, I ask whether available functional genomic evidence supports or contradicts the application of the theories of para-hormesis and xenohormesis to dietary natural phenols.” But beside listing overall effects of various genomic studies with resveratrol, curcumin (as in previous reviews) genistein, quercetin and luteolin, no “in-deep” analysis or mapping of relevant targets for these compounds are found in this review . There is some focus on genomic analysis showing effect on apoptosis, but this is not considered in the discussion.

3.       Abstract is not really focused. All the terms  “para-hormesis”, “xenohormesis”, ”holistic” and “multilevel” are used in the text and it is not easy to see the connection between functional genomic analysis and “para-hormesis” etc.

4.       Section 1, which is an introduction, is too overall to help understanding the result part as well as the discussion part. Terms like “para-hormesis”, “xenohormesis” is not described here. Further, the reader need to have an introduction to what the author means about ”holistic” and “multilevel”. I’m afraid that we not all agree on the meaning of these words.  And, why is it relevant to introduce a “holistic” and a “multilevel” view on the interaction of dietary bioactive compounds? I find that section 1 should be reduced to 1 page max and then focus of the subject the author want to address.

5.       As indicated at pkt 3, the connection between functional genomic analysis and “para-hormesis” etc. and therefore the review is pointing in different directions. Focus on of the themes and go to the end with these themes.

6.       Tabel 1 seems to be a mix of table 1 and what should have been Table 2 (not existing at all (only the heading)). What is the role of table 3? With no table legend, it is hard to understand the election of data presented in the tables.

7.       I feel that the author has not made a proper manuscript or it is only in a preliminary version, why I haven’t listed all the specific problems I find with the text and in the tables.

I haven’t listed all the specific challenges as the manuscript need to be significantly updated before I have to spent more time on commenting the paper. Sorry for my negative attitude.

Author Response

Referee 2.
>The subject of this review is very interesting and not very well addressed up to now: How is
>the combined effect of multiple dietary bioactive compounds on multiple targets in a complex >system as a cell? Therefore, I looked very much forward to review the manuscript, but I was >honestly disappointed when I realized that the paper is in a what I will call “a preliminary
>version”. 

In principle I agree with the referee that more work was needed to polish this text. I tried to re-focus the manuscript on multilevel chemopreventive effects. In particular, I removed any discussion of xeno-hormesis and para-hormesis. However, please, note that I intentionally focused on gene expression changes (rather than on biochemical interactions with receptors / enzymes).

>Major concerns
>1.       This is a follow-up on previous reviews from the same author concerning resveratrol
>and curcumin. It is mostly a repetition of what is already found in the previous reviews
>(Huminiecki, L. and J. Horbanczuk (2018). "The functional genomic studies of resveratrol in
>respect to its anti-cancer effects." Biotechnology advances 36(6): 1699-1708.
>And Huminiecki, L., J. Horbanczuk, et al. (2017). "The functional genomic studies of
>curcumin." Seminars in cancer biology 46: 107- 647). It is not clear whether new data are
>included. I don’t think so.

I scanned new literature and made sure that the review is up-to-date and relevant. I also considered genistein, quercetin, luteolin in comparison to resveratrol and curcumin. My aim is to provide an extensive theoretical synthesis rather than to repeat previous reviews.

>2.       It is stated that “my aim herein is to argue for a holistic and contextualized interpretation
>of all the genomic studies. In particular, I ask whether available functional genomic evidence >supports or contradicts the application of the theories of para-hormesis and xenohormesis to >dietary natural phenols.” But beside listing overall effects of various genomic studies with >resveratrol, curcumin (as in previous reviews) genistein, quercetin and luteolin, no “in-deep” >analysis or mapping of relevant targets for these compounds are found in this review . There
>is some focus on genomic analysis showing effect on apoptosis, but this is not considered in
>the discussion.

Genomic analyses are the focus of this review. I did not intent to provide any mapping to protein targets. I must underline this fact. Analyses of biochemical targets of natural phenols have been published dozens of times. Instead, I focused on gene expression changes!

>3.       Abstract is not really focused. All the terms  “para-hormesis”, “xenohormesis”, ”holistic”
>and “multilevel” are used in the text and it is not easy to see the connection between
>functional genomic analysis and “para-hormesis” etc.

I removed any mention of para-hormesis or xenohormesis from the manuscript. I removed the wordholistic as it fits more into literature in the field of philosophy of biology. Instead, I use the wordmultilevel that fits better in a molecular biology journal.

>4. Section 1, which is an introduction, is too overall to help understanding the result part
>as well as the discussion part. Terms like “para-hormesis”, “xenohormesis” is not described
>here. Further, the reader need to have an introduction to what the author means about
>”holistic” and “multilevel”. I’m afraid that we not all agree on the meaning of these
>words.  And, why is it relevant to introduce a “holistic” and a “multilevel” view on the
>interaction of dietary bioactive compounds?

I removed any mention of para-hormesis or xenohormesis from the manuscript. I removed the word ‘holistic’ that fits more into philosophy of biology literature. Instead, I use a word ‘multilevel’ that fits better in a molecular biology journal.

>I find that section 1 should be reduced to 1 page
>max and then focus of the subject the author want to address.

I think Introduction is reasonably structured and at reasonable length. I needed to introduce natural phenols, carcinogenesis and apoptosis as all later sections rely on these facts and concepts.

>5.       As indicated at pkt 3, the connection between functional genomic analysis and “para->hormesis” etc. and therefore the review is pointing in different directions. Focus on of the
>themes and go to the end with these themes.

I removed any mention of para-hormesis and xenohormesis from the manuscript. The focus is exclusively on functional genomic analyses.

​ 

>6.       Tabel 1 seems to be a mix of table 1 and what should have been Table 2 (not existing
>at all (only the heading)). What is the role of table 3? With no table legend, it is hard to
>understand the election of data presented in the tables.

There was a mistake in how these tables were formatted in the first version you got. Later, I asked the Editor to send out a correctly formatted manuscript. Please, also note that tables were re-arranged in the new submission. Further note that two new tables were added.

>7.       I feel that the author has not made a proper manuscript or it is only in a preliminary
>version, why I haven’t listed all the specific problems I find with the text and in the tables.
>I haven’t listed all the specific challenges as the manuscript need to be significantly updated
>before I have to spent more time on commenting the paper. Sorry for my negative attitude.

I disagree somewhat as the manuscript makes a number of novel but well-supported points. Underlining multilevel effects of natural phenols improves our understanding of how natural phenols act on the human body. Focus on expression changes serves this purpose well. This is an important area of interest to general readership. Natural phenols also have very significant practical potential in the chemoprevention of cancer.

Reviewer 3 Report

 1. The English needs to be checked and corrected by a native English writer.

2. Please improve introduction and conclusion.

3. References need to be past 6 years unless important.

4. The scheme of Apoptosis: programmed cell death, should be added in manuscript.

.

Author Response

Referee 3.

>1. The English needs to be checked and corrected by a native English writer.

I tried to improve the flow of the text. If there are any specific grammatical mistakes to correct, please, provide us with line numbers. (In practice, it is clarity and flow of the text that is most important for today’s busy readers.)

>2. Please improve introduction and conclusion.

I think Introduction is reasonably structured and at reasonable length. I need to introduce natural phenols, carcinogenesis and apoptosis as all later sections rely on these facts and concepts.

In the concluding section 3, I simply argue for the multilevel model. Again, I underline that there is no simple reductionist explanation for all anti-cancer effects of curcumin and resveratrol. The review makes this point very strongly, and somewhat against most papers published to-date (which focus either on sirtuins or Nrf2).

>3. References need to be past 6 years unless important.

I removed a number of less important references that were older than 6 years.

>4. The scheme of Apoptosis: programmed cell death, should be added in manuscript.

A scheme for apoptosis was added as Figure 3.

Reviewer 4 Report

Manuscript ijms-1929845  here is evidence for holistic anti-cancer activities of natural phenols in functional genomic studies of curcumin, resveratrol, genistein, quercetin, and luteolin by Lukasz Huminiecki* is review with data about functional genomic studies of natural phenols in the context of cancer.

Content is enough original and discussive and in my opinion will have interest of scientists. I have some question and recommendations for this manuscript.

1. Why did the author pay attention to these particular compounds among the huge number of natural phenolic compounds with anticancer potential?

2. Compounds chosen belong to different classes of phenolic compounds and discussing of their reactivity and affinity is usually discussed in scientific literature. In my opinion unferstanding of these properties is the base for any discussion of their action. 

3. To discuss chemopreventive mechanism of action chemical structures should be included and relevant biochemical pathways should be discussed. I think these data correlation with genomic studies result could explain better author’s hypothesis and its evidence.

Author Response

Referee 4.

>Manuscript ijms-1929845  here is evidence for holistic anti-cancer activities of natural
>phenols in functional genomic studies of curcumin, resveratrol, genistein, quercetin, and >luteolin by Lukasz Huminiecki* is review with data about functional genomic studies of
>natural phenols in the context of cancer.

>Content is enough original and discussive and in my opinion will have interest of scientists. I
>have some question and recommendations for this manuscript.

>1. Why did the author pay attention to these particular compounds among the huge number
>of natural phenolic compounds with anticancer potential?

I focused on representative small-molecule natural phenols important to cancer chemoprevention. In Table 1, I underlined the importance of these compounds in the light of their importance for cancer chemoprevention. There are thousands of relevant scientific articles and hundreds of reviews. There are dozens of relevant clinical trials and hundreds of food supplements.

>2. Compounds chosen belong to different classes of phenolic compounds and discussing of >their reactivity and affinity is usually discussed in scientific literature. In my opinion
>unferstanding of these properties is the base for any discussion of their action. 

Chemical properties of natural phenols are discussed in depth on page 4 in paragraph 1. More than a page is devoted to the discussion of free radical chemistry, nucleophilic addition, and Michael addition. Examples are given. Keap1 and the Nrf2 pathway are mentioned.

>3. To discuss chemopreventive mechanism of action chemical structures should be >included and relevant biochemical pathways should be discussed. I think these data
>correlation with genomic studies result could explain better author’s hypothesis and its
>evidence.

Chemical structures of curcumin, resveratrol, genistein, quercetin, and luteolin are shown in Figure 2.